# A Novel Carboxylesterase Derived from a Compost Metagenome Exhibiting High Stability and Activity towards High Salinity

**DOI:** 10.3390/genes12010122

**Published:** 2021-01-19

**Authors:** Mingji Lu, Rolf Daniel

**Affiliations:** Department of Genomic and Applied Microbiology and Göttingen Genomics Laboratory, Institute of Microbiology and Genetics, Georg-August-University of Göttingen, Grisebachstr. 8, 37077 Göttingen, Germany; mingji.lu@biologie.uni-goettingen.de

**Keywords:** carboxylesterases, metagenome, compost, lipolytic enzymes, halotolerance, halophilic, haloadaptation

## Abstract

Halotolerant lipolytic enzymes have gained growing interest, due to potential applications under harsh conditions, such as hypersalinity and presence of organic solvents. In this study, a lipolytic gene, *est56*, encoding 287 amino acids was identified by functional screening of a compost metagenome. Subsequently, the gene was heterologously expressed, and the recombinant protein (Est56) was purified and characterized. Est56 is a mesophilic (T_opt_ 50 °C) and moderate alkaliphilic (pH_opt_ 8) enzyme, showing high thermostability at 30 and 40 °C. Strikingly, Est56 is halotolerant as it exhibited high activity and stability in the presence of up to 4 M NaCl or KCl. Est56 also displayed enhanced stability against high temperatures (50 and 60 °C) and urea (2, 4, and 6 M) in the presence of NaCl. In addition, the recently reported halotolerant lipolytic enzymes were summarized. Phylogenetic analysis grouped these enzymes into 13 lipolytic protein families. The majority (45%) including Est56 belonged to family IV. To explore the haloadaptation of halotolerant enzymes, the amino acid composition between halotolerant and halophilic enzymes was statistically compared. The most distinctive feature of halophilic from non-halophilic enzymes are the higher content of acidic residues (Asp and Glu), and a lower content of lysine, aliphatic hydrophobic (Leu, Met and Ile) and polar (Asn) residues. The amino acid composition and 3-D structure analysis suggested that the high content of acidic residues (Asp and Glu, 12.2%) and low content of lysine residues (0.7%), as well as the excess of surface-exposed acidic residues might be responsible for the haloadaptation of Est56.

## 1. Introduction

Halophilic and halotolerant enzymes, which show resistance to salinity, are one of the major groups of extremozymes with industrial relevance. Microorganisms growing optimally at high salt concentrations are reservoirs for halophilic and halotolerant enzymes. They are generally divided into halophiles and halotolerant organisms [1,2]. To survive in high salinity, osmotic balance between cell cytoplasm and the external medium has to be maintained [3]. Halophilic archaea of the *Halobacteriales* order primarily adopt the “salt-in” strategy by accumulating equimolar concentrations of inorganic ions such as potassium and chloride ions [4,5]. This mechanism of osmoregulation results in intracellular enzymes, which evolve to halophilic types [6,7], which intrinsically show high activity and stability towards increasing salinity. In contrast, halophilic methanogenic archaea, as well as most halophilic and halotolerant bacteria largely employ the “salt-out” strategy by excluding salt from the cell inside and synthesizing and/or accumulating compatible organic osmolytes [8]. Some show also a combination of “salt-in” and “salt-out” strategies [9,10]. Halotolerant enzymes identified from “salt-out” microorganisms usually show different levels of salinity resistance [11,12,13]. Unlike most halophilic enzymes, which are inactive under low salt concentrations, halotolerant enzymes are still active in the absence of NaCl [14].

Haloadaptation of halophilic enzymes has been extensively studied by amino acid sequence and 3D structure comparison [15,16,17]. The presence of an unusually high proportion of acid residues and a drastic reduction of lysine residues on the surface of proteins play a key role in haloadaptation of enzymes [18,19]. Comparative analyses of halotolerant and halophilic enzymes with respect to amino acid compositions or 3D structures are rare. Given that scrutiny of protein and genome sequences may not unravel small differences during sequence-based comparative analysis [2,20], homologous enzymes sharing relatively high sequence similarity and conserved 3D structure would be promising in unveiling differences potentially related to shared strategies of adaptation to high salt environments.

Lipolytic enzymes, including esterases (EC 3.1.1.1, carboxylesterase) and true lipases (EC 3.1.1.3, triacylglycerol acyl hydrolase), are involved in catalyzing the cleavage and formation of ester bonds. Esterases prefer short-chain substrates with an acyl chain length of less than 10 carbon atoms, while lipases mainly catalyze the hydrolysis of long-chain triacylglycerols (≥10 carbon atoms) [21]. A distinguishing feature of lipolytic enzyme sequences is the conserved catalytic triad composed of a serine residue, which is located in the GXSXG consensus sequence, an aspartate or glutamate, and a histidine residue [22]. Most lipolytic enzymes also exhibit a similar core topology that typically consists of parallel β-pleated strands connected by α-helices [21]. Based on amino acid sequences and biological properties of lipolytic enzymes, 19 families (family I-XIX) have been identified [23]. Subsequently, new families such as EstLiu [24], Em3L4 [25], FLS18 [26], and Est9x [27] were added. Halophilic and halotolerant lipolytic enzymes have been detected in different families but the majority belonged to family IV [11,28,29,30]. Family IV esterases share high amino acid sequence similarity with mammalian hormone-sensitive lipases (HSL) and hence have also been referred to as the HSL family [21]. Previous studies on members of this family have predominantly explored the thermostability [31,32,33] and substrate specificity [34,35]. However, studies on salt tolerance of HSL family members are often missing.

Halophilic lipolytic enzymes are rare, among them, only LipC [36], MGS-B1 [37], and LipS2 [38] were characterized. In contrast, an increasing number of halotolerant lipolytic enzymes have been identified, through culture-dependent [28,39] and metagenomic approaches [29,40,41,42]. Apart from general features, such as broad substrate spectrum, chemo-, regio-, and enantio-selectivity and nonrequirement of cofactors, halophilic and halotolerant enzymes also tend to be resistant to organic solvents [15,31,43]. High salinity or presence of organic solvents both result in reduced water activity. Thus, halophilic and halotolerant lipolytic enzymes are advantageous in applications involving nonaqueous and aqueous/organic media, such as the degradation of organic pollutants in saline wastewater, bioremediation of oil spills, production of inter-esterification substances in food industry and nonaqueous synthesis of non-natural chemical compounds [14,44,45].

Composting is a process of decomposition and humification of organic matter [46]. During composting, enzymes secreted by microorganisms play key roles in degrading various organic substances. Recently, extremophilic enzymes, such as thermophilic, organic solvent tolerant, and alkaliphilic lipolytic enzymes, have been successfully identified from compost metagenomes [47,48,49]. However, there are no current reports of halophilic or halotolerant lipolytic enzymes from compost. In our previous study, a compost sample at the thermophilic stage (55 °C) was used to construct a metagenomic plasmid library [50] of which the identified esterase-encoding gene *est56* was selected for characterization. Est56 was a novel member of lipolytic family IV and further characterized as halotolerant. The haloadaptation mechanism of Est56 was also explored by comparing its amino acid composition with other halotolerant and halophilic enzymes, and subsequent analysis of homology-modelled 3D structures.

## 2. Materials and Methods

### 2.1. Bacterial Strains and Plasmids

The lipolytic recombinant plasmid (pFLD56) harboring the putative lipolytic gene *est56* was derived from the functional screen of the metagenomic compost library as described in our previous study [50]. The plasmid pFLD56 was used as template for amplification of *est56*. *E. coli* strain BL21(DE3) and the plasmid pET101/D-TOPO^®^ (Invitrogen, Karlsruhe, Germany) were used as expression host and vector, respectively.

### 2.2. Identification and Analysis of Est56 Sequence

Putative open reading frames (ORF) encoding lipolytic enzymes were initially predicted using the ORF Finder program (http://www.ncbi.nlm.nih.gov/gorf/gorf.html), and further verified using FramePlot analysis [51]. Potential signal peptides were detected using SignalP 4.0 [52]. Amino acid sequence similarity searches were performed using the BLASTP program against the public GenBank database [53]. Amino acid sequences of esterases homologous to the deduced gene product of est56 were aligned by employing clustalW. Secondary structure prediction was performed with I-TASSER [54]. Annotation of aligned sequences was performed with ESPript 3.0 [55].

### 2.3. Cloning and Expression of Est56

The putative lipolytic gene *est56* was amplified using the following primers: 5′-CACCATGCTCGCGCAGTCAC-3′ and 5′-CCCCTGGCGCGGGTAGTGTTCG-3′. The est56 PCR product was cloned into pET101/D-TOPO^®^ vector (Invitrogen, Germany), following the manufacturer’s instructions. The resulting recombinant plasmid was transformed into *E. coli* BL21 (DE3) cells. For expression of *est56*, a 6 mL preculture, incubated overnight, was used to inoculate 600 mL LB medium containing 100 µg/mL ampicillin. The culture was incubated overnight with shaking at 30 °C to an OD_600_ of 0.6. Expression was induced by adding IPTG (isopropyl-b-D thiogalactopyranoside) to a final concentration of 0.5 mM. After incubation for 6 h at 30 °C, cells were harvested by centrifugation (7000× *g*, 10 min, 4 °C), suspended in lysis-equilibration-wash buffer (50 mM NaH_2_PO_4_, 300 mM NaCl, pH 8), and then lysed on ice by sonication using a UPS200S homogenizer (Hielscher Ultrasonics GmbH, Teltow, Germany).

### 2.4. Purification of Recombinant Est56

To purify His_6_-tagged Est56, Protino^®^ Ni-TED 2000 packed columns (Macherey-Nagel, Düren, Germany) were used as recommended by the manufacturer. SDS-PAGE was performed to determine the purity and molecular mass of Est56. Protein concentration was measured by the Bradford method [56]. Finally, fractions containing the purified enzyme were pooled and dialyzed against 50 mM sodium phosphate buffer (pH 8) at 4 °C.

### 2.5. Esterase Standard Assay

Unless otherwise mentioned, Est56 activity was measured in 1.0 mL assay buffer containing 50 mM sodium phosphate (pH 8) and 1 mM *p*-nitrophenyl (*p*-NP) butyrate (Sigma-Aldrich, Munich, Germany), at 50 °C. The amount of *p*-NP released by enzyme-catalyzed hydrolysis was continuously monitored for at least 2 min at a wavelength of 410 nm against an enzyme-free reference. One unit of enzymatic activity was defined as the amount of Est56 needed to release 1 μmol of *p*-NP per minute under the assay conditions. All experiments were performed in at least triplicate, and extinction coefficients of *p*-NP under every assay condition were determined. Results are shown as mean values ± standard deviation (SD).

### 2.6. Characterization of Est56

#### 2.6.1. Substrate Specificity

Substrate specificity of Est56 was assessed under standard assay conditions using 1 mM *p*-NP esters (Sigma-Aldrich, Munich, Germany) of different chain lengths as substrates: *p*-NP acetate (C2), *p*-NP butyrate (C4), *p*-NP valerate (C5), *p*-NP caproate (C6), *p*-NP caprylate (C8), *p*-NP caprate (C10), *p*-NP laurate (C12), *p*-NP myristate (C14), and *p*-NP palmitate (C16). All substrates were prepared as a 0.1 M stock solution dissolved in isopropanol. For long chain substrates (>C10), the stock solution was first heated (50 °C) for a short time until the formation of clear transparent solution [40]. Initial reaction rates were calculated by estimating Est56 activity with different substrate concentrations ranging from 5 to 5000 µM. Michaelis–Menten constant (K_m_) and the maximal velocity (V_max_) were determined by employing Lineweaver–Burk plots [57].

#### 2.6.2. Effect of Temperature and pH

Optimum temperature of Est56 activity was determined between 20 and 70 °C. To assess protein thermostability, Est56 was preincubated in assay buffer at 30, 40, 50, and 60 °C for different time periods, and subsequently, residual activity was measured under standard assay condition.

Due to pH-dependent absorption of p-NP in acidic buffers [58], the effect of pH on Est56 activity was determined between pH values 3 and 10 at 348 nm (the pH-independent isosbestic wavelength). The overlapping buffer systems used comprised 50 mM acetate buffer (pH 3.0 to 6.0), 50 mM sodium phosphate buffer (pH 6.0 to 8.0), 50 mM TAPS (3-(2,4 dinitrostyrl)-(6*R*,7*R*-7-(2-thienylacetamido)-ceph-3-em-4-carboxylic acid) buffer (pH 8.0–9.0), and 50 mM CHES (N-cyclohexyl-2-aminoethanesulfonic acid) buffer (pH 9.0 to 10.0). The effect of pH on Est56 stability was examined by preincubating the enzyme at the respective pH value at 10 °C for 24 h. Subsequently, residual activity was measured under standard assay condition.

#### 2.6.3. Effect of Salinity

The effect of salinity on Est56 activity was measured by adding 0.5 to 4 M NaCl or KCl to the standard reaction assay mixture. Est56 stability against salt was determined by incubating the enzyme in assay buffer containing NaCl (0.5 to 4 M) at 10 °C for 24 h. Residual activity was measured under standard assay conditions.

Salt has been reported to protect halophilic proteins against denaturants (such as high temperature and urea) [18]. In this study, the protective effect of NaCl was investigated by adding different amounts of NaCl (0–4 M) to the incubation buffer, in which Est56 was incubated at different temperatures or in the presence of different amounts of urea. Specifically, Est56 was incubated at high temperatures (50 or 60 °C) for 30 min. For urea impact, Est56 was incubated with different amounts of urea (2, 4, and 6 M) at 10 °C for 24 h. Each additive was equalized to the same final concentration in the assay buffer, and the residual activity was measured under standard assay conditions. A blank reference was prepared using the buffer solution without enzyme but containing the same amount and type of additive. Activity measured before incubation was taken as 100%.

#### 2.6.4. Effect of Organic Solvents

The effect of organic solvents on Est56 stability was assayed by incubating Est56 in the presence of 15% and 30% (*v*/*v*) water-miscible organic solvents (DMSO, methanol, ethanol, acetone, isopropanol, and 1-propanol) or water-immiscible organic solvents (ethyl acetate, diethyl ether, chloroform, and toluene) at 10 °C for 24 h under constant shaking. Residual activity was measured under standard assay conditions.

#### 2.6.5. Effect of Other Additives

The effect of metal ions including K^+^, Ca^2+^, Mn^2+^, Mg^2+^, Zn^2+^, Fe^2+^, Cu^2+^, Ni^2+^, Fe^3+^, and Al^3+^ and inhibitors including phenylmethyl-sulfonyl fluoride (PMSF), dithiothreitol (DTT), and ethylenediaminetetraacetic acid (EDTA) was measured at concentrations of 1 and 10 mM. In addition, the impact of detergents such as Triton X-100, Tween 20, Tween 80, and SDS at concentrations of 0.1%, 1% and 5% (*v*/*v*) on enzyme activity was investigated. The catalytic activity of Est56 was measured under standard reaction conditions by directly adding each additive to the standard assay mixture. Activity measured in additive-free assay buffer was regarded as 100% activity, while reactions that included corresponding additive but no enzyme were used as blanks.

### 2.7. Sequence Analysis of Halotolerant Enzymes

Phylogenetic trees were constructed with the neighbor-joining method using MEGA version 7 [59]. For this purpose, Est56, other reported halotolerant, and reference lipolytic enzyme sequences retrieved from GenBank were employed (Appendix A). A bootstrap value of 500 replicates was used to estimate the confidence level [60]. The phylogenetic tree was subsequently visualized by GraPhlAn [61]. In addition, a tree containing only family IV esterases was constructed.

Characterized halophilic lipolytic enzymes are rare but enzymes from halophilic microorganism that adopt the “salt-in” strategy were evolved to be halophilic [6]. Thus, we retrieved 22 putative halophilic lipolytic enzymes originating from archaea of the Halobacteriales order from GenBank (Appendix A). The halophilic feature of each archaeal strain was checked on HaloDom webserver (http://www.halodom.bio.auth.gr) [62]. We summarized experimentally confirmed halophilic proteins as controls (Appendix A). Thereafter, we used HT_Lip, HP_Lip, and HP_Enz to refer to the 40 halotolerant lipolytic enzymes, 22 putative halophilic lipolytic enzymes, and 16 experimentally confirmed halophilic proteins, respectively. Amino acid composition, theoretical pI values, and molecular weight of each protein were calculated by the ProtParam tool at Expasy (www.expasy.org) [63].

Differences in amino acid compositions among HT_Lip, HP_Lip, and HP_Enz were statistically compared. Individual amino acids were analyzed as variants for each enzyme. Nonparametric Kruskal–Wallis (KW) and Mann–Whitney (MW) pairwise post hoc tests were used to evaluate median differences among univariate groups. Analysis of similarities (ANOSIM) test was used to pairwise compare the overall differences between multivariate groups based on Bray–Curtis distance, with 9999 permutations. A high R value generated by the ANOSIM test indicates a high dissimilarity between groups. Similarity percentage (SIMPER) tool calculates the average contribution of individual amino acids to the average dissimilarity between groups based on Bray–Curtis similarity. Statistical analyses were performed with R (http://www.r-project.org) using the “vegan” package [64].

### 2.8. Homology Modeling and Putative Structure Analysis

Based on deduced amino acid sequence, the tertiary structure prediction of Est56 was performed by I-TASSER [54]. PyMOL (PyMOL molecular graphics system, DeLano Scientific, Palo Alto, CA, USA; http://www.pymol.org) was used to visualize the predicted model. The analog with the highest TM score was also selected for structural superimposition. The surface electrostatic potential was calculated by the APBS plugin [65] and visualized by PyMOL.

### 2.9. Accession Numbers

The amino acid sequence of Est56 is available in the GenBank database under accession number KR149569.1. The compost metagenome sequences are available in the NCBI sequence read archive (SRA) under the accession number SRR13115019.

## 3. Results

### 3.1. Identification and Sequence Analysis of a Novel Lipolytic Gene

A metagenomic plasmid library derived from compost using *Escherichia coli* as host was constructed and function-based screened for genes conferring lipolytic activity as previously described [50]. An *E. coli* clone showing strong lipolytic activity (large halos) on indicator plates was selected for further characterization. Sequence analysis of the plasmid insert (6.1 kb, Appendix A) revealed a putative lipolytic gene (*est56*, 864 bp) encoding 287 amino acids. Putative signal peptides indicating extracytoplasmic localization were not detected in the deduced protein sequence. Similarity searches showed that protein sequences similar to Est56 were mainly identified during metagenome screenings. These comprised ELP45 isolated from a forest topsoil [66], EstC23 from mountain soil [67], Est06 from forest soil [68], and EstMY from activated sludge [69], which showed 62%, 61%, 60%, and 60% amino acid identity to Est56, respectively.

Multiple sequence alignments of Est56 with other esterases revealed that Est56 belongs to the HSL group (family IV) of lipolytic proteins. The conserved family IV motif H-G-G was present in the Est56 protein at amino acid sequence positions from 62 to 64. This motif plays an essential role in the stabilization of the oxyanion hole and catalysis. The catalytic triad composed of Ser^132^, Glu^226^, and His^256^, as well as another conserved motif E-X-L-X-D-D (amino acid residues from 226 to 231), was also detected in Est56 amino acid sequence (Appendix A).

### 3.2. Purification of Recombinant Est56 and Substrate Specificity

Est56 was heterologously produced in *E. coli* BL21 (DE3). After purification by Ni-TED affinity chromatography, Est56 was purified 73-fold with a specific activity of 90.44 U/mg (Appendix A). SDS-PAGE revealed a single band with a molecular mass of approximately 34.0 kDa (Appendix A). This is consistent with the calculated protein mass including the sequences for the V5 epitope and His_6_-tag, which were added during cloning of *est56*.

Assays with *p*-NP esters showed that Est56 exhibited a substrate preference for esters with short-chain fatty acids such as *p*-NP acetate (C2), *p*-NP butyrate (C4), *p*-NP valerate (C5), and *p*-NP caproate (C6). The enzyme did show little or no significant activity by employing *p*-NP esters with long-chain fatty acids (C8–C16) as substrates (Appendix A). This indicates that Est56 is an esterase and not a lipase [21]. The maximal specific activity was detected with *p*-NP butyrate (C4). The K_m_ and V_max_ values with this substrate were 128.0 µM and 102.0 U/mg, respectively.

### 3.3. Effect of Temperature and pH

Est56 retained high activity (above 40%) over the entire tested temperature range from 20 to 70 °C with maximal activity at 50 °C (Figure 1a). Despite the high activity (above 80%) of Est56 at 50 and 60 °C, thermostability at 50 and 60 °C was low. Est56 retained 50% activity after 30 min of incubation at 50 °C and 12.4% after 10 min at 60 °C. However, Est56 showed high stability for extended incubation times at 30 and 40 °C, with a half-life of 192 and 16 h, respectively (Figure 1c).

Est56 exhibited more than 80% activity between pH 6 and 8, with maximal activity at pH 8 (Figure 1b). Similarly, Est56 was most stable from pH 6 to 8, retaining more than 90% residual activity after 24 h incubation at 10 °C (Figure 1d).

### 3.4. Effect of Salinity

Addition of salt (NaCl or KCl) produced a stimulatory effect on Est56 activity, with an enhanced activity at NaCl and KCl concentrations of up to 2.5 M. Maximal activities compared to the reference without addition were recorded in the presence of 1.5 M NaCl (130.4%) and 1 M KCl (141.6%). At higher concentrations, Est56 activity decreased gradually with increasing concentration of NaCl and KCl. Notably, Est56 still retained approximately 90% activity at 3 M NaCl and 3.5 M KCl and 40% activity at 4 M NaCl and KCl (Figure 2a).

As for the stability, Est56 was stable over the tested NaCl and KCl concentration ranges (0–4 M), with almost unaltered activity after 24 h incubation at 10 °C (Figure 2b).

### 3.5. Protective Effect of NaCl against Denaturants

The presence of NaCl enhanced stability of Est56 to temperature (Table 1). At 50 °C, the addition of 1, 2, and 3 M NaCl significantly (*p* < 0.05) increased Est56 residual activity (above 70%) compared to that incubated in the salt-free assay. The best stabilization effect was detected at 2 M NaCl. However, incubation with 4 M NaCl significantly (*p* < 0.05) decreased Est56 stability. At 60 °C, Est56 was almost inactivated after 30-min without NaCl addition, whereas a significant increase in residual activity was detected by adding 1, 2, 3, or 4 M NaCl.

Urea is another denaturant that can cause inactivation of an enzyme directly and indirectly [70]. Addition of 3 M NaCl significantly (*p* < 0.05) enhanced Est56 tolerance against 2 M urea, with the residual activity increasing from 36.3% without NaCl addition to 54.0% in the presence of 3 M NaCl. A stabilizing effect of NaCl (*p* < 0.05) on Est56 activity was also observed in the presence of 4 or 6 M urea (Table 1).

### 3.6. Effect of Organic Solvents

Generally, Est56 exhibited enhanced activity after incubating with 15% and 30% (*v*/*v*) water-miscible organic solvents (Table 2), with the exception of 30% (*v*/*v*) 1-propanol, which caused a considerable loss of Est56 residual activity (23.0%). The highest stimulation of residual activity (286%) was observed after incubation with 30% (*v*/*v*) isopropanol. In the presence of water-immiscible organic solvents, Est56 was inhibited by 15% and 30% (*v*/*v*) ethyl acetate, chloroform, and toluene (Table 3). Est56 retained its full activity after incubation with 15% and 30% (*v*/*v*) diethyl ether.

### 3.7. Effect of Other Additives

Metal ions exhibited different effects on Est56 activity (Appendix A). The addition of Al^3+^ and Ca^2+^ at 1 and 10 mM had a stimulatory effect (approximately 130%). Est56 retained its full activity in the presence of 1 and 10 mM Mg^2+^. In contrast, Est56 activity slightly decreased to approximately 80% in the presence of 1 mM Fe^2+^, 10 mM Mn^2+^, and 1 and 10 mM Zn^2+^. The additives Cu^2+^ and Ni^2+^ at 1 and 10 mM and Fe^2+^ and Fe^3+^ at 10 mM were deleterious to Est56, as enzyme activity dropped to approximately 20%. EDTA did not affect Est56 activity, which indicated that Est56 activity is independent of divalent cations.

The nonionic detergents Triton X-100, Tween 20, and Tween 80 at 0.1% (*v*/*v*) significantly enhanced Est56 activity to 156.8%, 154.3%, and 112.7%, respectively. Est56 activity was inhibited or inactivated in the presence of 1% and 5% (*v*/*v*) of the other tested detergents (Appendix A). The addition of the inhibitors DTT, PMSF, and DEPC at 1 and 10 mM had detrimental effects on Est56 activity (Appendix A). The inhibition of enzyme activity by PMSF and DEPC indicated that serine and histidine residues, respectively, are part of the Est56 catalytic triad [71].

### 3.8. Sequence Analysis of Halotolerant Lipolytic Enzymes

A phylogenetic tree was constructed to group 40 halotolerant lipolytic enzymes (Appendix A) into families based on Arpigny and Jaeger [21] classification. As shown in Figure 3, these enzymes covered nine Arpigny and Jaeger families including family I, II, IV, V, VI, VII, VIII, XV, and XVII, as well as four new families including Est9x [27], EstLiu [24], lp_3505 [72], and EM3L4 [25]. Most of the analyzed halotolerant enzymes (18 enzymes) including Est56 belonged to Family IV. Family IV can be further divided into two sub-families based on the conserved GXSXG motif [34]. Est56 and other 10 halotolerant lipolytic enzymes belong to the GDSAG motif subfamily and the remaining 7 enzymes grouped into the GTSAG motif subfamily (Appendix A).

Generally, the overall amino acid compositions among 3 different groups of selected and characterized halotolerant or halophilic enzymes were analyzed. The halotolerant lipolytic enzymes (HT_Lip; Appendix A), the halophilic lipolytic enzymes (HP_Lip; Appendix A), and other characterized halophilic enzymes (HP_Enz; Appendix A) are pairwise different (ANOSIM test, *p* < 0.01 in all three cases). The R values between HT_Lip and HP_Lip (0.4254) and HT_Lip and HP_Enz (0.4251) were higher than that of HP_Lip and HP_Enz (0.2065). This result indicated a high separation of amino acid composition between halotolerant (HT_Lip) and halophilic (HP_Lip and HP_Enz) enzymes, rather than within halophilic enzymes. This result was also consistent with the average dissimilarity revealed by SIMPER analysis (Appendix A). The KW test identified that 12 residues (Asp, Glu, Lys, Arg, His, Leu, Met, Ile, Ala, Ser, Asn, and Gln) (Figure 4a) and theoretical pI values (Figure 4b) were significantly different (*p* < 0.05) among the three groups. The groups of halophilic enzymes HP_Lip and HP_Enz obtained significantly higher content of aspartic acid, glutamic acid, and arginine residues and lower content of lysine, leucine, methionine, isoleucine, and asparagine residues and theoretical pI values than those in halotolerant lipolytic enzymes HT_Lip (MW post hoc pairwise test, *p* < 0.05). As reported above, the most notable feature for the halophilic adaptation of halophilic enzymes is the excessive number of acidic residues (Asp and Glu) compared to lysine residues. This is reflected by low theoretical pI values [15]. However, halotolerant lipolytic enzymes exhibited broad range of theoretical pI values (4.59 to 9.44) (Figure 4b). Among them, Est56 obtains high content of acidic residues (Asp and Glu, 12.12%) and particularly low content of lysine residues (0.7%), as well as a relatively low theoretical pI value of 4.97.

### 3.9. Structural Modeling of Est56

The tertiary structure of Est56 is composed of a cap domain and an α/β-hydrolase fold core domain. The cap domain of Est56 consists of α-helices at the N-terminal side (α1 to α2) and between β6 and β7 (α6 to α7) (Figure 5a). The core domain comprises six helices surrounded by eight β-strands that form parallel structures. The catalytic triad of Est56 consists of Ser^132^ located between β5 and α5, Glu^226^ after β7, and His^256^ between β8 and α9 (Figure 5a). The overall structure of Est56 superimposed well (TM-score 0.984; RMSD 0.48) on E40 [73], with a global amino acid sequence identity of 53% (Figure 5b). The electrostatic potential of Est56 was calculated and described. The distribution of charges revealed that Est56 had negative charges on the surface (Figure 5c,d).

## 4. Discussion

In this study, we identified a halotolerant esterase, Est56, from a compost metagenome and compared it to recently reported halotolerant enzymes (Appendix A). In total, the 46 halotolerant lipolytic enzymes were grouped into 13 lipolytic families, which suggested a high diversity of halotolerant lipolytic enzymes (Figure 3). Including Est56, 29 halotolerant lipolytic enzymes were derived from metagenomes and correspondingly from uncultured microorganisms, which confirms the efficiency of metagenomic approaches in exploring novel enzymes [74,75]. The remaining 17 halotolerant enzymes were derived from individual microorganisms of which most thrive in saline environments, i.e., *Zunongwangia profunda* was derived from a surface seawater [24], *Erythrobacter seohaensis* from a tidal flat [76], *Psychrobacter celer* from a deep-sea sediment [77], and *Alkalibacterium* sp. SL3 from a soda lake sediment [78]. Some of the metagenome-derived halotolerant enzymes also originated from saline environments, such as marine water [27,37,41,79], deep sea sponges [42,80], deep-sea sediments [25,81], and deep-sea shrimps [82]. Recently, compost was reported as an habitat for halophilic and halotolerant microorganisms [83,84]. To our knowledge, Est56 is the first halotolerant esterase identified from a compost metagenome.

Est56 activity exhibited a T_opt_ at 50 °C and high stability at 30 °C (Figure 2). This is also the case for the majority of halotolerant esterases, which show a temperature optimum of enzyme activity between 35 and 50 °C and high stability at low temperatures (below 30 °C in most cases) [34,85,86,87]. Nevertheless, psychrophilic and thermophilic halotolerant lipolytic enzymes were also detected. For example, 7N9 [42], lp_3505 [72], EstS [11], Est10 [88], and EstLiu [24] displayed a temperature optimum ranging from 0 to 30 °C, whereas E69 [76], Est9x [27], BmEST [89], and LipJ2 [39] exhibit their optimal activity between 60 and 80 °C. With respect to pH, most of the halotolerant enzymes including Est56 are moderate alkaliphilic, with the optimum pH between 7.5 and 9 (Appendix A).

Generally, halotolerant lipolytic enzymes could be classified into three groups, according to their NaCl-dependent catalytic activity (Appendix A). In the first group comprising Lpc53E1 [80], PE10 [86], Lip3 [90], MGS-RG1 [82], LipC12 [40], Est9x [27], and Est700 [91] NaCl stimulated esterase activity was recorded over the tested concentration range (0–4 M in most cases). The second group harbors enzymes for which esterase activity continuously decreases with increasing NaCl concentration. Members of this group are 7N9 [42], ABO_1251 [79], Esth [92], ABO_1197 [79], EstS [11], and EstLiu [24]. For the third group, NaCl serves as an activator at low or moderate concentrations and as inhibitor at higher concentrations. Est56 belonged to this group, as its activity increased in the presence of NaCl up to a concentration of 1.5 M and decreased at higher concentrations (Figure 2). Moreover, almost 50% of the described halotolerant esterases such as E69 [76], ThaEst2349 [85], E25 [34], lp_3505 [72], and Est10 [88] follow this pattern.

Evaluation of enzyme stability towards salts revealed that Est56 retained its full activity after 24 h of incubation with NaCl or KCl concentrations ranging from 0 to 4 M at 10 °C (Figure 2b). Similarly, almost unaltered residual activity was also detected for esterases such as Lpc53E1 [80], EaEST [93], Esth [92], EstSL3 [78], LipC12 [40], Est12 [77], Est-OKK [94], and Est700 [91]. Nevertheless, esterases such as EstSP [30], EstS [11], EstLiu [24], and H8 [81] were inhibited by high salt concentrations. Noteworthy are the stabilities of EM3L4 [25], ThaEst2349 [85], Lip3 [90], and Est10 [88], which were enhanced by addition of NaCl. Although potassium ions are preferable for some halophilic enzymes [15,95], NaCl and KCl exhibited similar effects on Est56 activity and stability. This was tested only for a few other enzymes and similar results were obtained [29].

To shed a light on the mechanism of how halotolerant lipolytic enzymes resist salts, we compared the amino acid compositions among the groups HT_Lip, HP_Lip, and HP_Enz (Figure 4). The amino acid composition between halophilic and halotolerant proteins are significantly different, although enzymes in all groups were salt resistant. Our results were generally consistent with previous comparisons between halophilic and nonhalophilic homologs [9,96,97,98,99]. The most crucial feature of halophilic enzymes (HP_Lip and HP_Enz) compared to halotolerant lipolytic enzymes (HT_Lip) is the higher content of acidic residues (Asp and Glu) accompanied by a lower content of lysine residues (Figure 4a). This feature was also reflected by the lower pI values of halophilic enzymes (Figure 4b). Another consistent feature for halophilic enzymes is the low hydrophobicity [4,15], which was presented by a low content of aliphalic hydrophobic residues (Leu, Met and Ile) in this study (Figure 4a). Significant difference in aromatic (Phe, Try, and Trp), small (Gly and Vla), and borderline (Ser and Thr) hydrophobic residues was not detected between HT_Lip and HP_Lip (Figure 4a). On the contrary, different results were reported for certain residues therein by comparing halophilic and non-halophilic homologs [97,99,100,101,102]. Moreover, a significantly lower content of polar amino acids (Gln and Asn) was recorded for HP_Lip compared to HT_Lip enzymes (Figure 4a), which has not been observed in other comparative analyses. In general, high acidity and low hydrophobicity reflected by the amino acid composition of halophilic enzymes is a distinctive feature for their “halophilic adaptation,” which simultaneously enables them to resist salinity. Nonetheless, we could not find clear patterns in amino acid composition among halotolerant lipolytic enzymes, except the broad range of theoretical pI values (Figure 4b).

Recently, Dassarma and Dassarma [2] proposed a correlation between the halophilic character and acidic nature of proteins by reviewing the proteomes of different halophilic and halotolerant bacteria. Further studies on halophilic protein structure also confirmed that excessive surface-exposed acidic residues are the basis for halophilic adaptation [5,103,104,105,106]. Thus, given the low pI values of halophilic enzymes, we assume that halotolerant lipolytic enzymes with relatively low pI values would follow a similar haloadaptation as halophilic enzymes. Est56 exhibits a low pI value (4.97), with predominantly acidic residues located at its surface (Figure 5c,d). This is an indication that Est56 applies a salt resistance strategy similar to that of halophilic enzymes. Moreover, the distinctive feature of enhanced Est56 stability against denaturants (high temperatures and urea) mediated by NaCl suggests that Est56 possess a halophilic character caused by the high acidic amino acid content. Another family IV esterase, ThaEst2349 (theoretical pI value 4.94), was also reported as halotolerant due to the high ratio of surface-exposed acidic residues [85]. Acidic amino acids were reported to have a greater capacity than other amino acids in keeping proteins hydralated, which is important for the solubility of protein under salt stress [95,107,108]. Interestingly, several reports on the haloadaptation of enzymes with higher pI values than Est56 indicated that instead of acidic residues, basic residues at protein surfaces played a key role in their halotolerance [91,109,110]. By site-directed mutagenesis, Zhang et al. [81] identified two basic residues Arg^195^ and Arg^236^ located on the surface of H8 (theoretical pI value 9.09), which were essential in the salt tolerance. Additional mechanisms were reported to contribute to the haloadaptation of halotolerant enzymes. By introducing hydrophobic residues in the cap and catalytic domain, the halotolerance of E40 (a family IV esterase) was significantly improved [31]. However, this adaptation hardly applies to all of the family IV esterases, since the cap domain is the most variable region [111]. The cap domain was even not observed for the family IV enzyme MGS-M1, which was also reported to resist high salinity [37]. Thus, the underlying mechanism for the haloadaptation of different lipolytic enzymes remains unclear.

To some extent, the hydration characteristics of halotolerant/halophilic enzymes may extend their function to nonaqueous environments [15,43]. In this study, Est56 was stable towards the tested water-miscible organic solvents (Table 2), and moderately tolerant to some water-immiscible organic solvents (Table 3). Some halotolerant lipolytic enzymes, such as LipC12 [40], estHIJ [112], EstSP [30], Est12 [77], and H8 [81] were also reported to resist the presence of organic solvents but to different degrees. The stability towards organic solvents could broaden Est56 application in organic solvent-mediated catalytic processes, such as flavor production in food industry, synthesis of antibiotics and anti-inflammatory compounds in pharmaceutical industry, and production of pesticides for agricultural applications [113,114,115,116].

Additionally, other properties, such as the effect of metal ions, inhibitors, and detergents on Est56 activity were also studied. Est56 activity was enhanced by Ca^2+^ and Al^3+^ at 1 and 10 mM. Several esterases and lipases are also reported to be activated by Ca^2+^ ions [117,118]. However, it is rare for esterases to show an increased activity in the presence of Al^3+^ ions. Est56 activity was enhanced at a low concentration (0.1%, *v*/*v*) of nonionic detergents such as Triton X-100, Tween 20, and Tween 80, while suppressed at high concentrations (1% and 5%, *v*/*v*). This similar concentration-dependent effect of detergents on esterases was also found in other studies [40,88,119].

## 5. Conclusions

A functional screening of a compost metagenome yielded an esterase, showing activity and stability over a salinity range of 0–4 M. The recently reported halotolerant lipolytic enzymes (40 in total) were also summarized and used for phylogenetic analysis in this study. To explore the haloadaptation of halotolerant lipolytic enzymes, their amino acid compositions were statistically compared with halophilic counterparts. However, no clear pattern was found in the amino acid composition in the halotolerant lipolytic enzymes. For Est56, the excessive content of acidic residues over lysine residues, as well as the predominantly negatively charged surface indicated that it applies a haloadaptation similar to that of halophilic enzymes. In addition, Est56 exhibits a tolerance toward various organic solvents and enhanced activity in the presence of Ca^2+^ and Al^3+^ ions and a low concentration (0.1%, *v*/*v*) of nonionic detergents. Thus, Est56 is a novel biocatalyst with application potential, particularly under high salinity and in nonaqueous environments.

## Figures and Tables

**Figure 1 genes-12-00122-f001:**
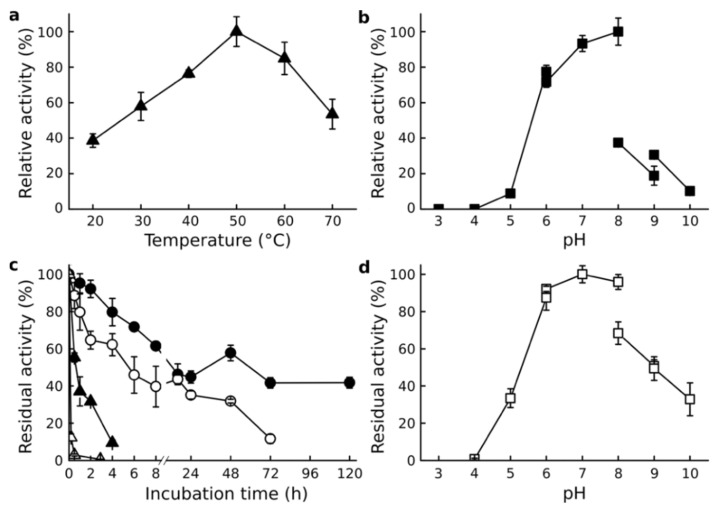
Effect of temperature and pH on Est56 activity. (**a**) Effect of temperature on Est56 activity; the maximal activity (153.9 U/mg) at 50 °C was taken as 100%. (**b**) Effect of pH on Est56 activity; the maximal activity (90.4 U/mg) at pH 8 was taken as 100%. (**c**) Thermostability of Est56 at 30 °C (closed circle), 40 °C (open circle), 50 °C (closed triangle), and 60 °C (open triangle); specific activity (87.2 U/mg) measured at the start of the incubation and under standard assay conditions was taken as 100%. (**d**) Effect of pH on Est56 stability was measured by incubating Est56 at 4 °C for 24 h, the maximal residual activity (33.1 U/mg) at pH 7 was taken as 100%.

**Figure 2 genes-12-00122-f002:**
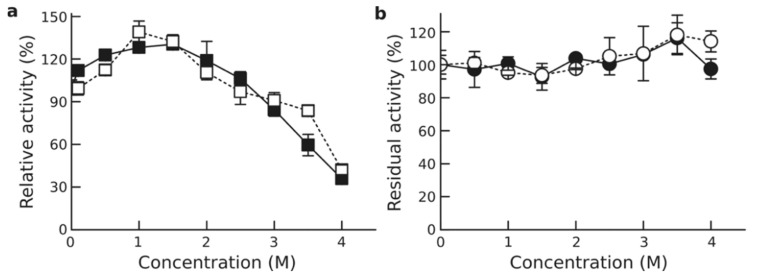
Effect of salinity on Est56 activity and stability. (**a**) Effect of NaCl (closed square) and KCl (open square) on Est56 activity. (**b**) Effect of NaCl (closed circle) and KCl (open circle) on Est56 stability. Specific activities corresponding to 100% activity were 93.1 and 94.1 U/mg for graph (**a**) and (**b**), respectively.

**Figure 3 genes-12-00122-f003:**
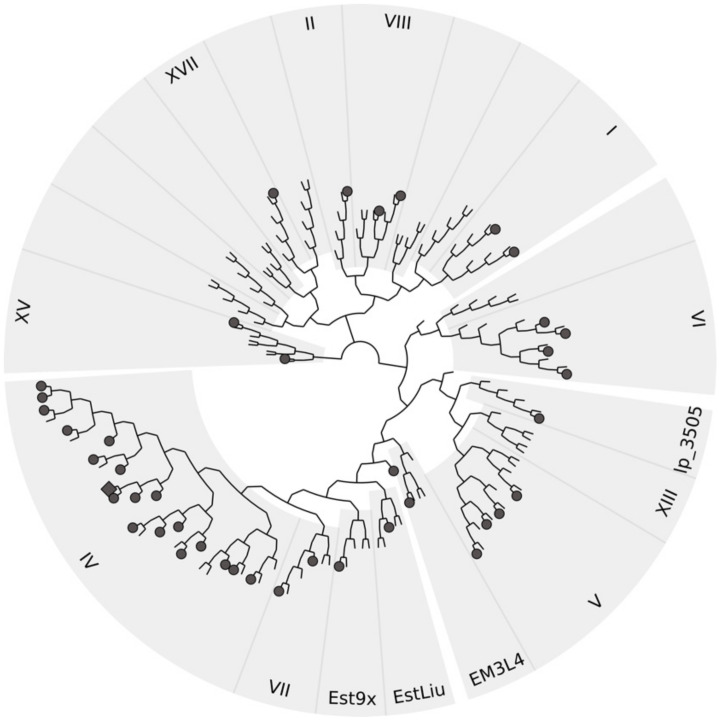
Phylogenetic classification of Est56 (closed diamond) and reported halotolerant (closed circle) lipolytic enzymes by neighbor-joining method. With the exception of Est56, other sequences were retrieved from GenBank.

**Figure 4 genes-12-00122-f004:**
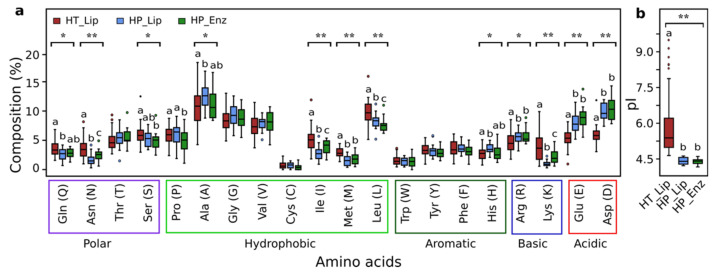
Comparison between halotolerant and halophilic enzymes. Box plots of (**a**) amino acid composition and (**b**) theoretical pI value of enzymes in groups HT_Lip (red), HP_Lip (blue), and HP_Enz (green). Statistical comparison was conducted by Kruskal–Wallis (KW) test (**, *p* < 0.01; *, *p* < 0.05) among three groups and Mann–Whitney (MW) post hoc test between pairwise groups (medians sharing a letter above boxes indicate no significant difference in the pairwise test).

**Figure 5 genes-12-00122-f005:**
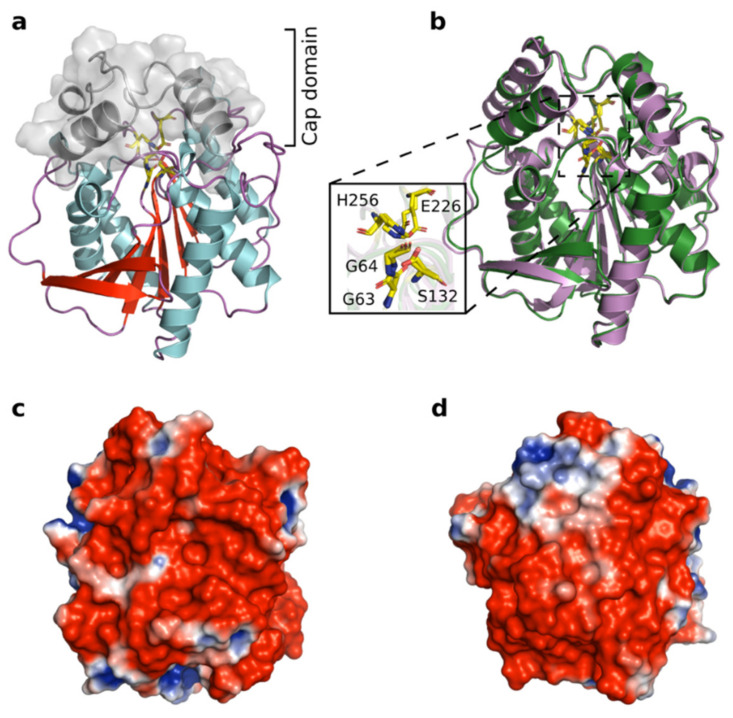
The modelled three-dimensional structure of Est56. (**a**) Ribbon representation of the Est56 monomer colored according to secondary structure elements. The overall structure is composed of two domains: cap domain and catalytic domain. (**b**) Superposition of Est56 (pink) onto its structural homolog E40 (PDB: 4xvc). The residues involved in stabilizing the oxyanion hole (Gly63 and Gly64) and catalytic triad (Ser94, Asp215 and His245) were indicated in stick representation (the sequence number is based on Est56 amino acid sequence). (**c**) Structural electrostatic potential of Est56. The most negative and most positive electrostatic potentials are indicated by red and blue, respectively, from −1 (red) to +1 kT/e (blue). (**d**) The 180° rotated view of (**c**).

**Table 1 genes-12-00122-t001:** Role of NaCl in protecting Est56 against denaturants.

	Residual Relative Activity (%) ^a^				
	0 M	1 M	2 M	3 M	4 M
**Temperature ^b^**					
**50 °C**	54.9 ± 2.4	71.3 ± 8.1 *	82.8 ± 7.4 **	72.9 ± 4.2 *	38.9 ± 3.1 **
**60 °C**	1.4 ± 0.7	12.7 ± 2.2 **	11.8 ± 1.6 **	8.0 ± 0.7 **	10.6 ± 0.6 **
**Urea ^c^**					
**2 M**	36.3 ± 4.7	36.2 ± 4.0	43.0 ± 4.7	54.0 ± 2.4 **	41.5 ± 9.5
**4 M**	19.5 ± 3.5	31.6 ± 2.5 **	33.6 ± 2.2 **	29.3 ± 4.0 *	ND ^d^
**6 M**	17.2 ± 2.4	28.6 ± 2.2 **	ND ^d^	ND ^d^	ND ^d^

^a^ Residual activity was measured after incubation with different amounts of NaCl. Two-sample *t* test was performed between residual activities incubated with and without NaCl. * α = 0.05; ** α = 0.01. ^b^ Est56 was incubated at 50 or 60 °C for 30 min in the presence of different concentrations of NaCl. Specific activity (41.9 U/mg) measured before incubation was taken as 100%. ^c^ Est56 was incubated at 10 °C for 24 h in the presence of different combinations of NaCl and urea. Specific activity (45.8 U/mg) measured before incubation was taken as 100%. ^d^ Not detectable.

**Table 2 genes-12-00122-t002:** Effect of water-miscible organic solvents on Est56 stability.

Organic Solvent	Residual Activity (%) ^a^	
	15% (*v/v*)	30% (*v*/*v*)
**DMSO**	135.0 ± 11.8	135.4 ± 4.7
**Methanol**	116.1 ± 6.7	96.6 ± 7.6
**Ethanol**	116.2 ± 8.7	107.3 ± 4.2
**Acetone**	138.3 ± 8.9	115.3 ± 14.8
**Isopropanol**	136.3 ± 10.1	286.6 ± 9.0
**1-Propanol**	114.0 ± 3.7	23.0 ± 1.0

^a^ Specific activity (33.4 U/mg) incubated in the organic solvent-free assay buffer was taken as 100%.

**Table 3 genes-12-00122-t003:** Effect of water-immiscible organic solvents on Est56 stability.

Organic Solvent	Residual Activity (%) ^a^	
	15% (*v*/*v*)	30% (*v*/*v*)
**Ethyl acetate**	55.3 ± 8.1	50.2 ± 3.5
**Diethyl ether**	110.8 ± 5.7	106.7 ± 10.8
**Chloroform**	46.5 ± 0.5	6.3 ± 2.5
**Toluene**	50.0 ± 2.2	54.9 ± 0.7

^a^ Specific activity (53.4 U/mg) incubated in the organic solvent-free assay buffer was taken as 100%.

## Data Availability

All datasets are publicly available stored in or derived from NCBI databases (https://www.ncbi.nlm.nih.gov). The amino acid sequence of Est56 is available in the GenBank database under accession number KR149569.1. The compost metagenome sequences are available in the NCBI sequence read archive (SRA) under the accession number SRR13115019.

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
