# Peer review of "A Novel Carboxylesterase Derived from a Compost Metagenome Exhibiting High Stability and Activity towards High Salinity"

_genes, 2021, doi:10.3390/genes12010122_

Round 1
Reviewer 1 Report
Manuscript contains new and significant information and is well and concise organized.
Objectives are clear.
Materials and methods are clear and complete.
Experiments are relevant to an international audience.
References are current, necessary, complete and appropriately cited in text.
Tables and figures with captions are clear and necessary.
The conclusions are supported by the data.
The results obtained by Mingji Lu and Rolf Daniel show evidently:
- The excessive content of acidic residues over lysine residues of Est56, as well as the predominantly negatively charged surface indicated that it applies a haloadaptation similar to that of halophilic enzymes.
- The Est56 exhibits a tolerance toward various organic solvents, and enhanced activity in the presence of Ca2+ and Al3+ ions and a low concentration (0.1%, v/v) of non-ionic detergents.
- Its amino acid composition and the homology-modelled 3D structures are correlated with other halotolerant and halophilic enzymes.
In conclusion, the esterase gene, est56, isolated from a compost metagenomic library, is a new halotolerant member of lipolytic family IV. The Est56 is a novel biocatalyst with application potential, particularly under high salinity and in non-aqueous environments.
Author Response
We thank the reviewer for his effort.
Reviewer 2 Report
The article aims to describe the characterization of a novel halotolerant carboxylesterase from a compost metagenome.
Novel halotolerant enzymes are needed for industrial and biotechnological applications, being capable to perform their reactions in presence of high concentrations of organic solvents.
The article is generally solid and describes a well-known pipeline, from the bioinformatic data to the biochemical enzymatic characterization. While lacking a proper crystallographic structure, the newfound enzyme is described with enough information regarding its activity optimal conditions (temperature, pH, salinity) and its resistance to urea and other additives that future biotechnological uses of this enzyme could be speculated nonetheless.
Furthermore, the phylogenetic analysis has been described indicating each of the taken steps (a not so obvious care), from the construction of the three databases to the statistical analysis of the amino acid compositions. The Discussion is undoubtedly a "wall of text", and Table S1, where the novel Est56 is associated with 45 other lipolytic halotolerant enzymes is surveyed for each of the previously analyzed enzymatic features of Est56.
Personally, for a more harmonious structure of the discussion I would move the comparison of Est56 with the other entries of the HT_Lip group regarding the stability with organic solvents and the effect of metal ions BEFORE the comparison of the amino acid sequences of HT_Lip with those of HP_Lip and HP_Enz. Following this order, the Discussion would end with the sentence "Thus, the underlying mechanism .... remains unclear", with a more intuitive link to the conclusion.
Further few more specific notes regard:
- Table 2-3: I am not sure that a line with Organic Solvent "None" is really needed (also because it creates a nonsense in regards to the subcolumns 15% (v / v) and 30% (v / v)). The a footnote is enough explainatory.
- The bacterial source of Est56 newfound enzyme is cited only in the Table S1. Even if the characteristics of the compost metagenome were previously described in ref. 50, the article could benefit of a brief phrase that summarize the previous work in the Introduction and of the metagenome accession number in M&M (if it is in open access).
Author Response
Furthermore, the phylogenetic analysis has been described indicating each of the taken steps (a not so obvious care), from the construction of the three databases to the statistical analysis of the amino acid compositions. The Discussion is undoubtedly a "wall of text", and Table S1, where the novel Est56 is associated with 45 other lipolytic halotolerant enzymes is surveyed for each of the previously analyzed enzymatic features of Est56.Personally, for a more harmonious structure of the discussion I would move the comparison of Est56 with the other entries of the HT_Lip group regarding the stability with organic solvents and the effect of metal ions BEFORE the comparison of the amino acid sequences of HT_Lip with those of HP_Lip and HP_Enz. Following this order, the Discussion would end with the sentence "Thus, the underlying mechanism .... remains unclear", with a more intuitive link to the conclusion.
Response: We want to keep the discussion in the original order. The main focus of this paper is to emphasize that Est56 is a salt-tolerant enzyme and then discuss its potential haloadaptation mechanism. Therefore, we put this part at the very beginning of the discussion. Moreover, we assumed that the resistance towards organic solvent of Est56 is an extension of its hydration characteristics. To support this, we mentioned several other tolerant lipolytic enzymes. However, the hydration characteristics of Est56 were discussed only after the haloadaptation mechanism discussion
Further few more specific notes regard:
- Table 2-3: I am not sure that a line with Organic Solvent "None" is really needed (also because it creates a nonsense in regards to the subcolumns 15% (v / v) and 30% (v / v)). The a footnote is enough explainatory.
We have corrected as the reviewer recommended. Please see the Table 2 and Table 3.
- The bacterial source of Est56 newfound enzyme is cited only in the Table S1. Even if the characteristics of the compost metagenome were previously described in ref. 50, the article could benefit of a brief phrase that summarize the previous work in the Introduction and of the metagenome accession number in M&M (if it is in open access).
We added some information in the introduction (lines 91-94). The accession number for the metagenome was also added, please see lines 547-548.
Reviewer 3 Report
Lu and Daniel are reporting a halotolerant esterase Est56 that they have obtained from a metagenomic plasmid library recruited from compost, selecting first by function-based screening and then by purification process using Escherichia coli as an expression host system. Entire manuscript is neatly written and many aspects of Est56 has been covered that can be used for downstream processing. The concept is not novel, however, given that the closest protein has ~60% amino acid sequence identity - Est56 is a welcoming member in the lipolytic family IV.
General comments:
Line 120 and 121 needs rephrasing: ……including the during cloning…..
Line 152, 197 and 199 contains abbreviation ‘approx.’. Please expand them to approximately.
Line 274, 275 and 276 needs rephrasing: ….enzymes (were) grouped…..
Specific comments:
Deduced Km value of 128.0 μM should belong to a high affinity enzyme (since you already tested Est56 activity over a wide range i.e. 5 to 5,000 μM (Line 439)), right? Could you corroborate whether it is an efficient or inefficient enzyme towards its natural substrate concentrations. Moreover, what is the natural substrate of Est56 in a compost system? Since you have already determined the absence of any signal peptide using SignalP program (Line 103, 104, 396, 397), Est56 is a cytoplasmic enzyme. Therefore, I assume that a variety of endogenous ester compounds should be accessible to Est56 intracellularly (similar to artificial substrates mentioned in lines 122-124). Is there any information available over this?
You have nicely summarized amino acid composition of halotolerant and halophilic type of enzymes in Table S8. Could you elaborate a bit more over thermostability factor? Looking over Est56 amino acid sequence reveals two cysteine residues. What are the likelihood of a cysteine=cysteine (disulfide bridge) formation? Could you highlight both cysteine residues in the PyMOL program and check if such bond formation is feasible in your modeling?
Furthermore, I like the information on the enzymes from microbes that were obtained from saline environments (Line 279-286). Normally, the salt concentration in brackish and seawater ranges between 15 - 35 g/L NaCl, respectively. What was the salt concentration in your compost system? If available, then add this information along with lines 279-286 or under haloadaptation part.
I would like to see an additional supplementary file that shows all the predicted genes (gene organization) on your 6.1-kB insert. It will reveal if this gene is a stand-alone candidate or part of an operon. Closest taxonomy relatedness of this protein is a Deltaproteobacteria bacterium (when you use BLASTP program). What happens if you conduct nucleotide homology search using complete 6.1-kB sequence? What are the closest bacteria you get as a hit? Additionally, if your insert has transposases or phage-related genes then it is a hint towards horizontal gene transfer amongst microbial communities in a compost microbiome.
What outgroup enzyme was used in your rooted phylogenetic tree? If not, then please add one and see if your tree gets modified!
Your finding that pertains to the stimulatory effect on Est56 activity by ~130% after the addition of Al3+ and Ca2+ attracts some additional clarifications. This is a nice finding from the application point of view. However, I cannot recall any aluminium specific transporter-encoding gene in a bacterial genome (to fuel e.g. cytosolic Est56). Only multidrug efflux transporters for the expulsion of toxic ions including heavy metals could be explained. Could you comment if upgraded catalysis by Al3+ (atomic number 13) addition has anything to do with its similarity to more physiologically relevant Mg2+ (atomic number 12) cofactor that anyways retains full enzymatic activity?
Author Response
Reviewer 3:
We thank the reviewer for the helpful comments to improve the manuscript. We paid regard as follows
Comments and Suggestions for Authors
Lu and Daniel are reporting a halotolerant esterase Est56 that they have obtained from a metagenomic plasmid library recruited from compost, selecting first by function-based screening and then by purification process using Escherichia coli as an expression host system. Entire manuscript is neatly written and many aspects of Est56 has been covered that can be used for downstream processing. The concept is not novel, however, given that the closest protein has ~60% amino acid sequence identity - Est56 is a welcoming member in the lipolytic family IV.
General comments:
Line 120 and 121 needs rephrasing: ……including the during cloning…..
Response. We have rephrased the part (lines 129-130)
Line 152, 197 and 199 contains abbreviation ‘approx.’. Please expand them to approximately.
Response: We have corrected as the reviewer recommended
Line 274, 275 and 276 needs rephrasing: ….enzymes (were) grouped…..
Response: We have corrected as the reviewer recommended (line 301)
Specific comments:
Deduced Km value of 128.0 μM should belong to a high affinity enzyme (since you already tested Est56 activity over a wide range i.e. 5 to 5,000 μM (Line 439)), right? Could you corroborate whether it is an efficient or inefficient enzyme towards its natural substrate concentrations. Moreover, what is the natural substrate of Est56 in a compost system? Since you have already determined the absence of any signal peptide using SignalP program (Line 103, 104, 396, 397), Est56 is a cytoplasmic enzyme. Therefore, I assume that a variety of endogenous ester compounds should be accessible to Est56 intracellularly (similar to artificial substrates mentioned in lines 122-124). Is there any information available over this?
Response: We don’t have information available on the natural substrate of Est56. But as the reviewer suggested, this could be one of the prospects for further study.
You have nicely summarized amino acid composition of halotolerant and halophilic type of enzymes in Table S8. Could you elaborate a bit more over thermostability factor? Looking over Est56 amino acid sequence reveals two cysteine residues. What are the likelihood of a cysteine=cysteine (disulfide bridge) formation? Could you highlight both cysteine residues in the PyMOL program and check if such bond formation is feasible in your modeling?
Response: As the reviewer suggested, we highlight the two cysteine residues (green sticks in the picture below) in the modelled Est56 3D structure. However, formation of the disulfide bond is not feasible. According to Figure 1, Est56 is a mesophilic protein, its thermostability is not outstanding. And in Table S9, the content of cysteine residues contributed least to the differences of amino acid composition among the three groups HT, HP_Lip, HP_Enz. Thus, in Figure 5, we only highlighted the features likely related to the haloadaptation of Est56 and did not include the cysteine residues.
Furthermore, I like the information on the enzymes from microbes that were obtained from saline environments (Line 279-286). Normally, the salt concentration in brackish and seawater ranges between 15 - 35 g/L NaCl, respectively. What was the salt concentration in your compost system? If available, then add this information along with lines 279-286 or under haloadaptation part.
Response: We have no information on the salt content of the compost.
I would like to see an additional supplementary file that shows all the predicted genes (gene organization) on your 6.1-kB insert. It will reveal if this gene is a stand-alone candidate or part of an operon. Closest taxonomy relatedness of this protein is a Deltaproteobacteria bacterium (when you use BLASTP program). What happens if you conduct nucleotide homology search using complete 6.1-kB sequence? What are the closest bacteria you get as a hit? Additionally, if your insert has transposases or phage-related genes then it is a hint towards horizontal gene transfer amongst microbial communities in a compost microbiome.
Resonse: We add another supplementary figure as the reviewer recommended. Please see new supplementary Figure S1. The insert doesn’t have putative transposases or phage-related genes. When we use the complete 6.1-kB sequence to conduct the nucleotide homology search, the closest bacterial hit is Tepidiforma bonchosmolovskayae but query coverage with only 40% and the identity were too low to draw solid conclusions, but we added the information of the organisms from which the best hits of the assigned ORFs located on the inserts were derived (Figure S1)
What outgroup enzyme was used in your rooted phylogenetic tree? If not, then please add one and see if your tree gets modified!
Response Normally, for new lipolytic enzymes classification, other studies use annotated lipolytic enzymes belonging in different families as references to construct a phylogenetic tree [1–3] without using a single specific enzyme as outgroup. In our case, Est56 belongs to Family IV (Figure 3). In Figure 3, only families containing halotolerant enzymes were marked with the corresponding family names. Thus, defining a single enzyme as outgroup taking the differences in the proteins into account is not really useful. Moreover, bootstrapping based on 500 resamplings was used to estimate the robustness of the tree in this study.
Your finding that pertains to the stimulatory effect on Est56 activity by ~130% after the addition of Al3+ and Ca2+ attracts some additional clarifications. This is a nice finding from the application point of view. However, I cannot recall any aluminium specific transporter-encoding gene in a bacterial genome (to fuel e.g. cytosolic Est56). Only multidrug efflux transporters for the expulsion of toxic ions including heavy metals could be explained. Could you comment if upgraded catalysis by Al3+ (atomic number 13) addition has anything to do with its similarity to more physiologically relevant Mg2+ (atomic number 12) cofactor that anyways retains full enzymatic activity?
Response: Metal ions influence enzyme activity by binding to different sites of enzyme. It iss common to observe the positive effected of Ca2+ on enzyme activity, due to the formation of a conserved calcium-binding by two conserved aspartic acid resides near the active-site [4,5]. For Al3+, as the reviewer mentioned, it is rare to observe the promoted effect on lipolytic enzyme activity. So far, we only found lipolytic enzymes, including lipases from Bacillus niacini EMB-5 [6], EstSHJ2 [7], est‐OKK[8], immobilized lipase [9] and OSTL28 [10], which show a slight activation by the addition of Al3+. However, none of these studies investigated the mechanism underlying the activation of the enzyme activity by Al3+. In addition, the correlations between the metal ions Al3+ and Mg2+ were not clear. Similar to our results, the enzyme activity of est‐OKK[8] and OSTL28 [10] was generally promoted by Al3+, while almost fully retained by Mg2+. Thus, the effect of metal ions on lipolytic enzymes seemed to be inconclusive, it related to factors such as metal ions, concentrations, and source of lipolytic enzymes. Thus, we cannot comment on the relation between Al3+ and Mg2+ activation.
References
- Kovacic, F.; Babic, N.; Krauss, U.; Jaeger, K. Classification of Lipolytic Enzymes from Bacteria. In Aerobic Utilization of Hydrocarbons, Oils and Lipids. Handbook of Hydrocarbon and Lipid Microbiology; Rojo, F., Ed.; Springer, Cham, 2019; pp. 1–35 ISBN 9783319397825.
- Biver, S.; Vandenbol, M. Characterization of three new carboxylic ester hydrolases isolated by functional screening of a forest soil metagenomic library. J. Ind. Microbiol. Biotechnol. 2013, 40, 191–200, doi:10.1007/s10295-012-1217-7.
- Ho Jeon, J.; Sook Lee, H.; Hun Lee, J.; Koo, B.-S.; Lee, C.-M.; Hee Lee, S.; Gyun Kang, S.; Lee, J.-H. A novel family VIII carboxylesterase hydrolysing third- and fourth-generation cephalosporins. Springerplus 2016, 5, 525, doi:10.1186/s40064-016-2172-y.
- Rashid, N.; Shimada, Y.; Ezaki, S.; Atomi, H.; Imanaka, T. Low-Temperature Lipase from Psychrotrophic Pseudomonas sp. Strain KB700A. Appl. Environ. Microbiol. 2001, 67, 4064–4069, doi:10.1128/AEM.67.9.4064-4069.2001.
- Sabri, S.; Rahman, R.N.Z.R.A.; Leow, T.C.; Basri, M.; Salleh, A.B. Secretory expression and characterization of a highly Ca2+-activated thermostable L2 lipase. Protein Expr. Purif. 2009, 68, 161–166, doi:10.1016/j.pep.2009.08.002.
- Oyedele, S.A.; Ayodeji, A.O.; Bamidele, O.S.; Ajele, J.O.; Fabunmi, T.B. Enhanced lipolytic activity potential of mutant Bacillus niacini EMB-5 Grown on Palm Oil Mill Effluent (POME) and biochemical characterization of purified lipase. Biocatal. Agric. Biotechnol. 2019, 18, 101017, doi:10.1016/j.bcab.2019.01.055.
- Wang, M.; Ai, L.; Zhang, M.; Wang, F.; Wang, C. Characterization of a novel halotolerant esterase from Chromohalobacter canadensis isolated from salt well mine. 3 Biotech 2020, 10, 430, doi:10.1007/s13205-020-02420-0.
- Yang, X.; Wu, L.; Xu, Y.; Ke, C.; Hu, F.; Xiao, X.; Huang, J. Identification and characterization of a novel alkalistable and salt-tolerant esterase from the deep-sea hydrothermal vent of the East Pacific Rise. Microbiology 2018, 7, e00601, doi:10.1002/mbo3.601.
- Chandel, C.; Kumar, A.; Kanwar, S.S. Enzymatic synthesis of butyl ferulate by silica-immobilized lipase in a non-aqueous medium. J. Biomater. Nanobiotechnol. 2011, 2, 400–408, doi:10.4236/jbnb.2011.24049.
- Fan, X.; Liu, X.; Wang, K.; Wang, S.; Huang, R.; Liu, Y. Highly soluble expression and molecular characterization of an organic solvent-stable and thermotolerant lipase originating from the metagenome. J. Mol. Catal. B Enzym. 2011, 72, 319–325, doi:10.1016/j.molcatb.2011.07.009.
